# Support Vector Machine-Based Formula for Detecting Suspected α Thalassemia Carriers: A Path toward Universal Screening

**DOI:** 10.3390/ijms25126446

**Published:** 2024-06-11

**Authors:** Idit Lachover-Roth, Sari Peretz, Hiba Zoabi, Eitam Harel, Leonid Livshits, Dvora Filon, Carina Levin, Ariel Koren

**Affiliations:** 1Allergy and Clinical Immunology Unit, Meir Medical Center, Kfar Saba 4428163, Israel; idit.roth@clalit.org.il; 2School of Medicine, Tel Aviv University, Tel Aviv 6997801, Israel; 3Hematology Laboratory, Emek Medical Center, Afula 1834111, Israel; sari_pe@clalit.org.il (S.P.); hiba.zoabi@gmail.com (H.Z.); eitam_ha@clalit.org.il (E.H.); 4Red Blood Cell Research Group, Vetsuisse Faculty, Institute of Veterinary Physiology, University of Zurich, 8057 Zürich, Switzerland; leonidlivshts@gmail.com; 5The Zurich Center for Integrative Human Physiology (ZIHP), 8057 Zürich, Switzerland; 6Hematology Division, Hadassah Medical Center, Jerusalem 9112001, Israel; filon@hadassah.org.il; 7Pediatric Hematology Unit, Research Laboratory, Emek Medical Center, Afula 1834111, Israel; levin_c@clalit.org.il; 8The Ruth and Bruce Rappaport Faculty of Medicine, Technion–Israel Institute of Technology, Haifa 3525422, Israel

**Keywords:** alpha thalassemia, screening, support vector machine, mathematical formulas

## Abstract

The blood counts of α thalassemia carriers (α-thal) are similar to those of β thalassemia carriers, except for Hemoglobin A_2_ (Hb A_2_), which is not elevated. The objective of this study was to determine whether mathematical formulas are effective for detecting suspected α-thal. The data were obtained from the database of the prevention program for detecting couples at risk for having a child with hemoglobinopathy. Red Blood Cells (RBC) indices were analyzed using mathematical formulas, and the sensitivity and negative predictive value (NPV) were calculated. Among 1334 blood counts suspected of α-thal analyzed, only the Shine and Lal and the Support Vector Machine formulas revealed high sensitivity and NPV. Sensitivity was 85.54 and 99.33%, and NPV was 98.93 and 99.93%, respectively. Molecular defects were found in 291, and 81 had normal α genes. Molecular analysis was not performed in 962 of the samples. Based on these results, mathematical formulas incorporating one of these reliable formulas for detecting suspected α or β thalassemia carriers in the program of the automatic analyzers can flag these results, increase the awareness of the primary physicians about the carrier risk, and send an alert with a recommendation for further testing.

## 1. Introduction

Hemoglobinopathies are the most common monogenic genetic diseases worldwide, including β and α thalassemia and Sickle Cell Disease (SCD). Each year, more than 300,000 new patients are born with these diseases, principally in developing countries in Africa and Asia [1]. Patients with β thalassemia and SCD experience severe symptoms that are difficult for the patients and their families and present a significant burden on healthcare services [2,3,4]. Therefore, prevention programs that can detect couples at risk of having an affected child have been implemented in many countries. Since both transfusion dependent (TDT) and non-transfusion dependent β thalassemia (NTDT) and SCD are generally more severe than α thalassemia, screening programs are conducted to detect carriers of those diseases [5,6]. Alpha gene deletions or additional mutations are much more frequent than β gene defects and more widely distributed, even before the extensive population migrations of the last centuries. About 5% of the world’s population carries an α gene defect [7].

In Asian countries, the incidence of α thalassemia carriers is much higher than the mean frequency worldwide. For example, in Vietnam, Myanmar, Laos, Thailand, and Cambodia, the frequency is 12% to 37% [8]. In Mediterranean countries, Africa, the Middle East, and the Indian subcontinent, α thalassemia carriers are frequent, but with a significantly lower prevalence than in southeastern Asian countries [9]. The relatively higher frequency of α thalassemia carriers in tropical and subtropical countries can be attributed to the selective advantage of α thalassemia carriers in areas where malaria is frequent, similar to the advantage of SCD and β thalassemia carriers [10]. In contrast to the β globin diseases, α thalassemia is characterized by the presence of two α genes on chromosome 16, the α1 and α2 genes (αα/αα). More than 100 α thalassemia genetic defects have been reported. The most common α gene defects are deletions, but other mutations are also known [7]. Clinically, there are four possibilities: four α intact genes (αα/αα), one affected gene that causes the silent α trait (-α/αα), two affected genes that cause the α thalassemia trait (-α/-α or (--/αα), three affected genes that cause the clinical form known as Hb H disease (--/-α), and four affected genes that cause the severe form of α thalassemia known as hydrops fetalis (--/--), which is usually lethal.

In the two cases of α thalassemia carriers, the silent and the α thalassemia trait, the blood count indices, mean corpuscular volume (MCV), and mean corpuscular hemoglobin (MCH) are low and in similar ranges but less affected than in carriers of β thalassemia. Usually, the Hb level can be slightly lower than normal, and the red blood cell (RBC) count is elevated compared to the Hb level. All carriers of α globin defects have normal high performance liquid chromatography (HPLC) analyses, or even low Hb A_2_ levels. The threshold for MCV is usually ≤78 fl, despite the fact that α and β thalassemia carriers can have MCV in the normal range of 80 fl or higher. For MCH, the usually accepted threshold is ≤27 pg. In some studies, the threshold is <80 fl for [11]. In a study that analyzed the RBC indices in 1301 α thalassemia carriers, the mean MCV was 74.36 ± 6.83 fl, with a wide range from 45 fl to even 88.8 fl. The MCH was also under 27 pg, 23.6 ± 2.7 pg, but also with a wide range from 13.1 to 30.5 pg [12]. Since the lower range for RBC and Hb in that study was 2.69 mm^3^ and 4.1 g/dL, respectively, some of the patients with low MCV and MCH likely had Hb H disease or α thalassemia carriers combined with iron deficiency anemia (IDA) and not only simple α thalassemia carriers.

The most common defects in the α genes are deletions, with the -α3.7 kb and -α4.2 kb deletions being the most common [7]. Non deletional mutations in the α genes can also cause the same clinical and laboratory pictures.

The genotype of the α carriers differs in eastern Asian countries, where the *cis* form (--/αα) is more common compared to the western world, where the *trans* form (-α/-α) is more common. Due to this difference, the frequency of hydrops fetalis and even the Hb H disease is common in Asian countries; therefore, routine screening for detection of α carriers is usually performed in Asian countries. Nonetheless, cases of Hb H disease and even hydrops fetalis have been described in western countries when only two or three mutations in the α gene were diagnosed (-α/-α) [7]. Non-deletional defects involving the α2 gene are more severe than deletional defects because the α2 gene encodes more α globin than does the α1 gene [13].

Interaction between α globin abnormal genes and β globin mutations can ameliorate or increase the severity of the clinical picture of α thalassemia carriers. For example, in cases of triplication or quadruplication of α thalassemia genes, this combination aggravates the severity of β thalassemia carriers, and in those subjects, a clinical picture of NTDT or even TDT can be present [14].

As mentioned, screening for β thalassemia and SCD is common in many countries, but a byproduct of the screening programs is the detection of possible α thalassemia carriers, where only molecular analysis can confirm the presence of an α gene defect. Since screening for α thalassemia carriers is not the major goal of those screening programs, the true incidence of α thalassemia carriers is unknown.

Rarely, β thalassemia carriers may have normal Hb A_2_ in HPLC analysis, called silent β carriers. in cases where both parents have RBC indices in the range to be considered suspected α or β globin carriers, even if one is a “classical β carrier”, and the other is defined as a “suspected carrier”, molecular analysis is recommended.

The principal differential diagnosis for low MCV and MCH is, of course, iron deficiency, but iron status is not routinely analyzed in most screening programs, and iron deficiency does not exclude the possibility of being a carrier of β or α thalassemia.

In the last few decades, investigators around the world have tried to use blood cell indices to develop a mathematical formula that would flag cases of suspected thalassemia carriers, mainly β thalassemia. Currently, none of these formulas have been adopted for routine use, mainly because there was not enough information regarding the ranges for α thalassemia carriers and the fear of false negative (FN) diagnoses in those cases.

The prevention program for detecting couples at risk of having a child with hemoglobinopathies was instituted in northern Israel in 1987 [6]. Since 2010, blood samples from women with red blood cell indices suspicious of being α thalassemia carriers have been sent for molecular analysis when the partner was also suspected of being a carrier.

The purpose of this study was to determine the impact of this screening approach and to propose more significant and cost-effective screening decisions based on the outcomes. We also tried to determine whether the formulas described in the literature are effective for detecting α thalassemia carriers among a homogenous population of women of childbearing age [15,16,17,18,19,20,21,22,23,24,25,26,27,28,29,30]. Some of these formulas were also analyzed in recent papers [31,32].

## 2. Results

A total of 22,842 blood count results were included in the data analysis, including the results published by our group when we compared the results of β thalassemia carriers with results that were defined as non_β thalassemia carriers [26]. In this paper, we analyzed the results of blood counts from 1334 women who were diagnosed or suspected to be α thalassemia carriers, as described in the Methods. In summary, 1334 results were α thalassemia carriers or suspected carriers (5.9%), 2936 were β thalassemia carriers (12.8%), and 18,572 had blood counts that did not correspond to α or β thalassemia carriers (81.3%; Table 1). The α thalassemia carriers had significantly lower RBC and higher Hb, MCV, MCH, MCHC, and RDW values (*p* < 0.001 for all) compared to the β thalassemia carriers. As expected, the A_2_ and Hb F levels in α thalassemia carriers were lower than in the β thalassemia group (*p* < 0.001 for both).

The group defined as α thalassemia carriers included 372 samples that were analyzed for molecular diagnosis. The results revealed 291 samples with mutations in the α globin gene; a normal α globin gene was found in 81 samples. Another 962 samples were defined as suspected α thalassemia carriers but did not undergo molecular analysis. Comparison of the blood count results from these three groups did not reveal significant differences except that the RBC were higher in the blood counts with proven mutations and lower in the group with normal sequences (*p* < 0.001). The differences in the RDW levels had a similar trend as the RBC results (*p* < 0.004; Table 2).

Due to the minimal differences between the three groups defined as α thalassemia carriers, normal molecular analysis, or suspected carriers, the analysis of the mathematical formula was combined as one α thalassemia group.

### 2.1. Comparative Results of the Formulas

Only the Shine and Lal formula [28], and our previously published SVM formula [26] had high sensitivity and NPV for detecting suspected α thalassemia carriers. The Shine and Lal formula had a sensitivity value of 85.54%, 88.22% specificity, and an NPV of 98.93%. The SVM formula had a sensitivity of 99.33%, 73.67% specificity, and 99.93% NPV. FN results were found in both the Shine and Lal and SVM formulas (n = 25,819.3%). Seventeen FN results detected in Shine and Lal’s formula were not detected in the SVM formula, and three results detected in the SVM formula were not detected by Shine and Lal’s formula.

Overall, 275 FN results were found when applying the SVM formula (20.6% of the 1334 samples), 42 FN results (18.1%) were found in the samples with a proven mutation (-α3.7 or IVS I-1 (-bp) 22.4% and 12.4%, respectively, *p* > 0.25), 22 FN samples were in the normal molecular analysis samples (27.2%), and 111/962 in the suspected α thalassemia carriers group (11.5%). When we compared the proportion of FN results in those groups, the *p*-Value was < 0.05.

The results of applying all the other formulas published in the literature failed to reveal sufficiently significant levels of sensitivity, all results were below 80%. NPV above 90% was found in several formulas, but with a sensitivity of <80%. Two formulas had 100% sensitivity but 0 NPV (Table 3).

### 2.2. Analysis of Samples Suspected to Have a Diagnosis of Iron Deficiency Anemia

Of 291 samples, 28 (9.62%) in the group with α genotype mutation or deletion had RBC ≤ 4.5 × 10^9^/dL and RDW ≥ 14.5%. Consequently, those samples can be defined as having IDA concomitant with an α thalassemia trait.

Of the 81 samples in a group with normal α gene sequencing, 20 (24.7%) had RBC ≤4.5 × 10^9^/dL and RDW ≥ 14.5% and can be defined as having IDA. In the group defined as having suspected α thalassemia (no molecular analysis), 111,962 samples (11.5%) had RBC ≤ 4.5 × 10^9^/dL and RDW ≥ 14.5%, which can then be defined as having IDA. The MCV and MCH values in all the α thalassemia groups were in the same range (*p* = NS).

### 2.3. Comparison of the Results from the Two Common α Globin Mutations Found

In 291 of 1334 blood counts, an α globin mutation was found (21.8%). The two most common deletions were -α3.7 kb and αIVS I-1 (-5 bp) (HBA_2_:c.95+2_95+6delTGAGG). In 134 samples (46%), the -α3.7 kb/αα genotype was found, and in another 24 (8.2%), the -α3.7 kb/-α3.7 kb genotype was detected. In 97 samples (33.3%), the αIVS I-1 (-5 bp)/αα genotype was found, and in another eight (2.7%), the genotype was αIVS I-1 (-5 bp)/αIVS I-1 (-5 bp). The genotype was αIVS I-1 (-5 bp)/-α3.7 kb in seven samples (2.4%). Another 25 genotypes were found, including α Poly A-HBA_2_:c.*92A>G in seven subjects, -α4.2 kb in five, Med in three, and others in five. We did not include the data analysis of the α CD 39—HBA:c.118-120delACC genotype commonly called Hb Taybe that presents with different clinical characteristics and was described elsewhere [34].

We compared the RBC indices for the two most common genotypes found in this study (Table 4).

## 3. Discussion

As hemoglobinopathies are the most common monogenic clinically severe disease worldwide, any effort toward detecting carriers and offering genetic counseling for couples at risk is recommended. Carriers of Sickle Cell Disease do not have any specific or characteristic features in RBC indices. Therefore, HPLC analysis is required to detect carriers. On the other hand, the indices of carriers of α or β thalassemia can be clearly identified and lead to suspicion of being carriers. Multiple efforts have ensued to find a reliable mathematical formula that is specific enough and has a high NPV to minimize further unnecessary laboratory analyses. A reliable formula will need to be tested on as many samples as possible. It can be incorporated into all laboratory blood analyzers and then flag results that need further analysis.

To date, no formula has been incorporated in any routinely used analyzers, despite publications that demonstrate the reliability of some formulas based on large samples. The likely cause is that HPLC analysis is also performed in standard, inexpensive analyzers, and most β thalassemia carriers can be detected. However, HPLC analysis is routinely done on request in screening programs, not in routine blood samples, and not for thalassemia screening. β thalassemia screening programs are routinely implemented in countries with many carriers, but not in all of them. In recent years, the migration of large numbers of people to countries where β thalassemia was not common has presented a new health problem that needs attention [2].

Detection of α thalassemia carriers is mostly relevant in populations of Asian origin, where the *cis* carriers are common (αα/--) and the risk of an infant with hydrops fetalis or severe forms of Hb H disease (α-/--) is frequent. Some *trans* carriers (α-/α-) with specific mutations in the α gene can also be found and produce an infant with severe α thalassemia disease. Mathematical formulas published in the literature do not usually include α thalassemia carriers in their analyses, except in two studies with a relatively small number of carriers: Amendiola et al. with 81 subjects and Mentzer [15,23] with 50 subjects. The current study is the largest study to detect α thalassemia carriers using mathematical formulas.

Since 1987, a systematic screening program for detecting couples at risk of having a child with β thalassemia has been instituted in northern Israel [5,6]. Since 2010, all samples with RBC indices suspicious for β or α thalassemia carriers were sent for molecular analysis, provided the partner was also suspected to be a carrier. Therefore, we have many blood count results that can be analyzed with mathematical formulas. The present study includes the analysis of three groups of blood counts and molecular analysis results. Since the RBC indices were similar in the three groups, especially MCV, MCH, and MCHC, we analyzed the results of the three groups together and compared the results with results previously published by our group, including β thalassemia carriers and normal samples [26].

The results indicate that only three published formulas have relatively high sensitivity: 68.63% in Ricerca et al. [24], 85.54% in Shine and Lal [28], and our previously described SVM formula with a sensitivity of 99.33% [26]. The respective NPVs for these formulas are 96.83%, 98.83%, and 99.93%. Two formulas used only the MCV and MCH values for calculation. Ricerca et al. [24]. used the RDW and RBC values. MCV and MCH values were also used by d’Onofrio et al. [33]. The sensitivity was 100%, but the NPV was 0. May be if the cutoff value were changed this formula would also become reliable, but we decided to compare the formulas using the same cutoff values that were used to detect β thalassemia, as shown in Table 3. Bordbar et al. also used the MCV and MCH indices; however, when we applied this formula to our results, the NPV was high enough, at 93.72%, but the sensitivity was only 21.46% [16]. Most of the published formulas that used different combinations of indices also had an NPV above 90% but a sensitivity below 60%. As shown in the meta-analysis published by Hoffmann et al. in 2015, indeed MCV/MCH are the most relevant parameters for discriminating thalassemia from IDA [35]. We also calculated the 75th and 95th percentiles of the formulas’ results and compared them with the published cutoff for each formula (Table 3). Our SVM formula passed the cutoff value for the 75th percentile but not the lower limit of the 95th percentile. None of the other formulas passed the cutoff value even for the 75th percentile, probably indicating that the cutoff is too high and some α thalassemia carriers might be missed. Recently, Phirom et al. described the use of machine learning in a small group of individuals with α thalassemia and showed that the Convolutional Neural Networks (CNN) formula is superior to the SVM formula that we use, but the specificity of the SVM formula, the formula that we apply for our study, is superior (81.23% vs. 84.94%), while the accuracy in their work is similar for both (CNN 80.69% vs. SVM 80.68%) [36].

We decided to include in our analysis only women and results with Hb > 9.0 g/dL to obtain a more homogeneous population for more accurate data analysis. Only 38 results (0.17%) were excluded due to Hb < 9.0 g/dL, and all except two samples were in the “suspected α carrier group”. The other two samples were in the normal molecular analysis group. In our opinion, excluding those samples did not affect the overall study results. Hb level < 9 g/dL may be due to IDA or even a combination of α carrier and IDA. Since the results analyzed in this study are the product of a massive screening program, iron status was not routinely analyzed.

The main differential diagnosis for microcytosis or hypochromia is, of course, iron deficiency, which could be present when the molecular analysis does not reveal alpha globin mutations. In all three groups of α thalassemia in the present study, 9.6% to 24.7% of the results can be indices suspicious for IDA, defined as RBC lower than 4.5 × 10^9^/dL and RDW above 14.5%. The higher percentage was from the group that did not undergo molecular analysis, which is reasonable. On the other hand, the low percentage of samples suspected to have IDA in the group with negative molecular analysis (11.5%) suggests that a mutation outside the α gene can produce characteristics of α thalassemia trait indices.

Indeed, some studies have described that individuals with characteristics of α thalassemia carriers may have intact α globin genes that may be functionally inactive. In these cases, large deletions upstream of the α globin genes, in the locus control region can cause inactivation of both α1 and α2 globin genes [37,38,39].

Also, deletions of the major regulatory element of the α globin genes that is responsible for activating the α globin genes can inactivate the normal α globin genes. In those cases, the MCV values are similar to those of classical α thalassemia carriers. Fifteen such deletions were described [40]. Other defects, not located in the α genes, are support nucleotide polymorphisms between the α-globin genes and the α gene regulatory element [40,41].

In summary, the fact that we did not find molecular defects in 6% of our molecularly analyzed samples did not preclude that they are not α thalassemia carriers. The similar RBC indices suggest that at least some might be α carriers.

We compared the RBC indices among the two most common genotypes in our study, -α3.7 kb and IVS I-1(-5 bp) (HBA_2_:c.95+2_95+6delTGAGG) (Table 4).

The values in the RBC indices were significantly different when we compared the one or two -α3.7 kb deletions and in the IVS I-1(-5 bp) genotypes, which were lower in the genotype with two deletions.

In a study from Iran that included 722 α thalassemia carriers, the same features were found, as carriers with two deletions in the -α3.7 kb gene (334 individuals) had significantly lower MCV and/or MCH values when compared to subjects with one deletion (45 subjects). (MCV 76.8 ± 5.02 fl vs. 72.7 ± 4.55 fl, respectively, and for MCH 24.7 ± 1.6 pg vs. 22.46 ± 1.73 pg, respectively [42].

The α-3.7 kb deletion is most prevalent in Mediterranean countries and in Africans. The Arab Bedouins in northern Israel are probably of African origin, which can explain the high incidence of this deletion found in the present study [43,44,45,46]. This is also the most common deletion found in Iran, in heterozygous or homozygous form (41.4% and 11.6%, respectively) [12].

As described, α thalassemia carriers are considered healthy subjects, but they still have significant microcytosis and mild anemia, principally in the pediatric population [47].

In general, prenatal screening for detecting couples at risk of having a child with α thalassemia in one or the two clinical forms is not cost-effective in countries outside of Asia, but in countries where screening programs for β thalassemia are implemented, including screening for α thalassemia, it does not add significant costs, and the decision to perform molecular analysis should be chosen, depending on the partner’s blood count indices and on the most frequent α thalassemia gene defects in each country [48].

The cost effectiveness of α thalassemia screening has not been estimated. However, β thalassemia screening was proven to be cost-effective [3].

For couples at risk of having an offspring with hydrops fetalis, genetic counseling should be offered, and the possible prenatal diagnosis discussed. Couples at risk of having an infant with Hb H should be informed, but prenatal diagnosis is not indicated since usually the course of the disease is mild with stable hemolytic anemia, rarely requiring blood transfusions. Also, couples at risk of having an offspring with two mutations in the α gene should be informed since, in the neonatal period, those newborns may have hemolysis requiring blood transfusions.

In couples at risk of having a child with Hb Taybe (α CD 39—HBA:c.118-120delACC [34] or other severe α gene defects such as Med or Poly A, hydrops fetalis can occur [49]. In those cases, genetic counseling and the option of prenatal diagnosis can be considered.

Based on our results, incorporating one of the formulas that we found to be significant for detecting suspected α or β thalassemia carriers in the program of the automatic analyzers, can flag the abnormal results and increase the awareness of primary physicians about the carrier risk, and if relevant, in cases of women of childbearing age, refer them for further analysis: HPLC for β thalassemia carriers and molecular analysis for α thalassemia carriers.

The major limitation of the present study is that iron status was not analyzed since the RBC indices and HPLC results were obtained as part of a very large-scale screening program for detecting carriers of hemoglobinopathies. Obviously, women with low Hb and/or RBC counts are suspected to have at least IDA or a combination of IDA and thalassemia carriers, which is relatively frequent in pregnant women.

This study was conducted only on women of childbearing age, and we cannot conclude whether those formulas are applicable to males or children. Further studies are needed to validate the results in other populations. At a minimum, even if the first stage of the formula is applied to this specific population only, it will be effective for population screening.

## 4. Methods

A prevention program for detecting pregnant women who were carriers of β thalassemia was initiated in the northern region of Israel from 1987. In 1999, the program was extended to include sickle cell carriers due to the relatively high frequency of sickle cell carriers in this area [6]. Through the end of 2023, more than 100,000 subjects were screened, and 2346 pregnancies at risk for having an affected child were detected and referred for genetic counseling.

Since 2010, all samples from women with low MCV (≤78) and/or low MCH (≤27), even with normal Hb A_2_ levels in HPLC analysis, were requested to provide blood samples from their partners. If the partners were found to carry a hemoglobinopathy, diagnosed by HPLC, or suspected to be carriers of α thalassemia based on the same MCV and MCH parameters but normal Hb A_2_, the blood samples were routinely referred for molecular analysis to detect or exclude α thalassemia carriers. If the partner’s RBC indices and the HPLC analysis were normal, the investigation of the couple was terminated, and there was no risk of having a child with a severe or clinically significant hemoglobinopathy. Otherwise, the samples were sent for further molecular analysis.

### 4.1. Red Blood Count Analysis

For the data analysis, we included the results of RBC and HPLC analyses of pregnant women or women of childbearing age to obtain a more homogeneous population. We excluded 39 blood counts (2.9%) with an Hb level less than 9 g/dL because the possibility of IDA or even a combination of α thalassemia carriers and IDA, which is more common among pregnant women, was likely.

The complete blood count analysis was performed on an ADVIA i2120 analyzer (SIEMENS, Forchheim, Germany, 91301). The Hb fractionation was analyzed using the HPLC Bio-Rad Variant-II set-up (Bio-Rad, Hercules, CA, USA).

### 4.2. Molecular Analysis

Alpha globin gene deletions were detected using the GAP-PCR methodology. For each of the common gene deletions: -α3.7 kb, -α4.2 kb, --Mediterranean, and -α20.5 kb, a set of specific primers is used. Point mutations are detected using Sanger sequencing following PCR amplification of each of the two alpha-globin genes. In rare cases, we used MLPA: Multiplex Ligation-Dependent Probe Amplification technology to detect deletions or multiplications.

According to the results of the molecular analysis, three groups were defined. The first group had an α thalassemia trait with proven molecular diagnosis; a second group included samples suspected to be α thalassemia carriers, but without proven mutations in the α globin gene detected; and a third group had suspected α thalassemia carriers but without molecular analysis. The third group included all the samples from pregnant or childbearing age women suspected to be α thalassemia carriers, whose partners had normal RBC indices and normal HPLC.

Subsequently, the results of the blood counts of the three groups of α thalassemia carriers were compared to the results of women in the same age range not suspected to be α or β thalassemia carriers and with the results of the women diagnosed with β thalassemia trait based on blood indices and HPLC analysis. The results of blood counts from non-carrier women and β thalassemia carriers were obtained from the same database of the prevention program, as described in a previous paper from our group [26].

### 4.3. Analysis of the Data Using Mathematical Formulas

The results of the three α thalassemia groups were also analyzed using the mathematical formulas described in the literature for detecting β thalassemia trait and the differential diagnosis from normal samples and/or iron deficiency to demonstrate the reliability of those formulas for detecting α thalassemia trait or carriers. A small number of samples of α globin carriers were included in two of the studies [15,23]

### 4.4. The Support Vector Machine (SVM) Algorithm

After analyzing the data with the published formulas, we applied another mathematical method based on the SVM algorithm to the same data. The SVM method that was published by our group for detecting blood counts suspected to be from β thalassemia carriers compared to normal counts or suspected of stemming from women with IDA. Details of this algorithm have been published elsewhere [50]. The SVM algorithm initially used all the relevant data from the RBC counts, including Hb, Hct, RBC, MCV, MCH, MCHC, and RDW. After checking the entire database, the algorithm chooses the most relevant values and discards the rest. We applied the previous SVM method for analyzing our data from α thalassemia carriers in the three groups described above and compared the results of the samples in the same database used in our previous study [26].
$$\frac{1.449994*\left(\mathrm{MCV}-82.80762\right)}{10.28125}+\frac{0.662983*\left(\mathrm{MCH}-27.02007\right)}{3.904478}+0.981605$$

### 4.5. Analysis of the Red Blood Count Indices and HPLC Results in the Two Most Common α Gene Defects

Finally, the results of a group of proven α thalassemia trait carriers with the two most common identified deletions found in our study, -3.7 kb and IVS I-1 (-5 bp) (HBA2:c.95+2_95+6delTGAGG), were analyzed. The heterozygous results were compared to the homozygous results for each deletion, and the common combinations between those deletions were analyzed.

Samples with the mutation α CD 39–HBA:c.118-120delACC, commonly called Hb Taybe, were excluded from the analysis because they presented a different, specific clinical picture and blood count ranges [34].

Since the blood samples and HPLC analyses were part of a very large-scale screening program, iron or ferritin levels were not analyzed.

### 4.6. Data Analysis

A students *t*-test or chi square was used for paired comparisons, and two-sided analysis of variance tests, assuming equal variances, were used for group comparisons. A *p*-Value of < 0.05 was considered significant. All analyses were performed using the SPSS statistical program version 28.

### 4.7. Ethics

The study was approved by the Emek Medical Center Ethics Committee, approval number EMC-52109-06, and registered at the NIH Clinical trials registry—NCT00481221.

## 5. Conclusions

Using mathematical formulas, especially the new generation of artificial intelligence, represented in this study by the Support Vector Machine (SVM)-developed formula, can detect individuals suspicious for being α thalassemia carriers, similar to the results of applying these formulas for detecting suspected β thalassemia carriers, as was demonstrated previously by our group [26].

This formula can and should be incorporated into the program of the automatic blood count analyzers to send an alert to the referring physician with a recommendation for further diagnostic tests, HPLC, and molecular analysis in the case of α thalassemia. By doing this, a universal screening program can be implemented without added costs, and the subsequent analysis can avoid the birth of affected children in specific circumstances, in the case of α thalassemia. Using the SVM formula can be beneficial since this formula can initially be trained on different groups of subjects, such as different genders, age groups, and different cutoffs used in different countries.

After this suggestion is implemented, additional large data studies can be conducted, with subsequent reevaluation and correction of the SVM formula, if necessary.

## Figures and Tables

**Figure 1 ijms-25-06446-f001:**
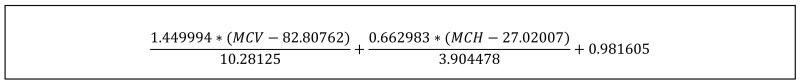
The Support Vector Machine Algorithm calculated formula [26]. MCV: mean corpuscular volume; MCH: mean corpuscular hemoglobin.

**Table 1 ijms-25-06446-t001:** Data from non-β thalassemia trait subjects, β thalassemia trait, and suspected α thalassemia trait.

Diagnosis	Non β Thalassemia Trait Mean ± SD (Range)	β Thalassemia Trait Mean ± SD (Range)	α Thalassemia Trait and/or Suspected α Trait Mean ± SD (Range)	*p*-Value between α and β Trait
Number of individuals (%)	18,572 (81.3)	2936 (12.8)	1334 (5.9)	
RBC (×10^9^/dL)	4.2 ± 0.43 (2.17–7.67)	5.42 ± 0.55 (3.21–7.81)	4.87 ± 0.44 (3.48–6.34)	<0.001
Hb (g/dL)	11.75 ± 1.06 (9.00–19.2)	10.65 ± 0.95 (9.00–15.4)	11.51 ± 1.07 (9.0–15.6)	<0.001
MCV (fl)	85.9 ± 6.84 (34–125.9)	63.14 ± 5.76 (48–91.5)	73.56 ± 4.52 (53.3–91.4)	<0.001
MCH (pg)	28.16 ± 2.7 (16.2–40.7)	19.71 ± 1.95 (14–31.3)	23.71 ± 1.92 (16.5–29.9)	<0.001
MCHC (g/dL)	32.75 ± 1.76 (12.6–45.7)	31.23 ± 1.73 (17.5–65)	32.17 ± 1.68 (19.2–36.2)	<0.001
RDW (%)	14.96 ± 2.02 (10.1–36.4)	16.39 ± 2.71 (12–22.8)	15.16 ± 1.94 (12.1–28)	<0.001
Hb F (%)	0.34 ± 0.61 (0–14)	2.12 ± 2.67 (0–38)	0.4 ± 0.29 (0.1–1.9)	<0.001
Hb A_2_ (%)	1.12 ± 1.38 (0–3.4)	5.6 ± 0.8 (3.5–8.8)	2.6 ± 0.26 (2–3.3)	<0.001

SD: standard deviation; RBC: red blood cells count; Hb: hemoglobin; MCV: mean corpuscular volume; MCH: mean corpuscular hemoglobin; MCHC: mean corpuscular hemoglobin concentration; RDW: red density width; Hb F: fetal hemoglobin; Hb A_2_: hemoglobin A_2_.

**Table 2 ijms-25-06446-t002:** Data from α thalassemia trait with proven genetic defect, suspected α thalassemia trait not molecularly analyzed, and suspected α thalassemia trait with normal molecular analysis.

Diagnosis	α Thalassemia Trait + Mutation Mean ± SD (Range)	α Thalassemia Trait Suspected Mean ± SD (Range)	α Thalassemia Trait Normal Sequence Mean ± SD (Range)	*p*-Value (*)
Number of individuals (%)	291 (21.8%)	962 (72.12%)	81 (6.07%)	
RBC (×10^9^/dL)	4.93 ± 0.46 (3.79–6.34)	4.87 ± 0.43 (3.48–6.27)	4.67 ± 0.36 (3.7–5.69)	<0.001
Hb (g/dL)	11.58 ± 0.97 (9.0–14.8)	11.51 ± 1.1 (9.0–15.6)	11.39 ± 0.93 (9.1–13.5)	NS
MCV (fl)	72.95 ± 5.17 (53.3–83.9)	73.55 ± 4.25 (57.8–91.4)	75.1 ± 4.07 (62.2–85)	NS
MCH (pg)	23.61 ± 2.1 (16.5–27.1)	23.66 ± 1.84 (16.5–29.9)	24.43 ± 1.81 (18.3–28.6)	NS
MCHC (g/dL)	32.26 ± 1.72 (19.2–36.10)	32.15 ± 1.33 (25.9–36.2)	32.51 ± 1.38 (29.4–35.5)	NS
RDW (%)	14.91 ± 1.8 (12.3–23.4)	15.19 ± 1.87 (12.1–28.0)	15.89 ± 2.35 (12.9–26.5)	0.004
Hb F (%)	0.5 ± 0.52 (0.1–7.3)	0.4 ± 0.29 (0.1–1.9)	0.4 ± 0.34 (0–5.1)	NS
Hb A_2_ (%)	2.6 ± 0.29 (1.3–3.6)	2.6 ± 0.26 (2–3.3)	2.6 ± 0.27 (0.8–3.3)	NS

SD: Standard deviation; RBC: red blood cells count; Hb: hemoglobin; MCV: mean corpuscular volume; MCH: mean corpuscular hemoglobin; MCHC: mean corpuscular hemoglobin concentration; RDW: red density width; Hb F: fetal hemoglobin; Hb A_2_: hemoglobin A_2_. (*) *p*-Value compared the three parameters.

**Table 3 ijms-25-06446-t003:** Analysis of the results of applying published mathematical formulas to our suspected α thalassemia carriers results. The published cutoff for β thalassemia was used.

No.	Study (Reference)	Formula	βThal Cut-Off	α Thal PPV	α Thal NPV	α Thal Specificity	α Thal Sensitivity	Percentile 75%	Percentile 95% (Lower Limit)
1	Srivastava [27]	MCH/RBC	<3.8	39.39	**93.59**	**99.35**	5.82	5.38	6.04 (5.97)
2	(1) England and Fraser, (2) England and Fraser (1979) [18,19]	MCV-RBC-(5-Hb)-K^*^	<0	0.87	**93.10**	**97.53**	0.3	70.29	73.37 (72.96)
3	Mentzer [23]	MCV/RBC	<13	45.72	**93.94**	**99.01**	11.57	16.45	18.24 (18.09)
4	Shine and Lal [28]	MCV^2^ × MCH/100	<1530	34.42	**98.83**	88.22	85.54	1466.58	1626.52 (1607.81)
5	Ricerca et al. [24]	RDW/RBC	<3.3	13.66	**96.83**	68.81	68.63	3.38	4.01 (3.95)
6	Green and King [21]	MCV^2^ × RDW/(Hb × 100)	<65	58.45	**95.15**	**98.44**	30.47	78.26	92.63 (90.12)
7	D’Onofrio et al. [33]	MCV/MCH	>0.9	6.74	0	0	**100**	3.18	3.34 (3.33)
8	Romero Artaza et al. [25]	RDW × MCV/RBC	<220	53.65	**95.86**	**97.4**	41.64	251.18	290.99 (285.58)
9	Sirdah et al. [30]	MCV-RBC-(3XHb)	<27	56.63	**93.69**	**99.55**	7.15	37.19	41.73 (41.11)
10	Ehsani et al. [17]	MCV-(10 × RBC)	<15	45.45	**93.86**	**99.1**	10.42	29.8	35.4 (34.8)
11	Sirachainan et al. [29]	1.5 × Hb–0.05 × MCV	<14	6.08	**91.95**	32.69	60.39	14.58	16.01 (15.85)
12	Bordbar et al. [16]	[80-MCV]*[27-MCH]	>44.76	9.17	**93.72**	84.63	21.46	37.8	101.32 (93.44)
13	[20]	Hb × RDW × 100/RBC^2^ × MCHC	<21	63.44	**95.2**	**98.72**	30.89	25.14	29.3 (28.66)
14	Hisham Index [22]	MCH × RDW/RBC	<67	55.85	**94.99**	**98.4**	28.13	80.87	94.52 (92.59)
15	Hameed Index [22]	MCH × Hct × RDW/(RBC × Hb)^2^	<220	6.73	0	0	**100**	4.99	6.52 (6.35)
16	Amendolia et al.—SVM [15]	SVM—(RBC, Hb, Hct, MCV)		98.75	13.59	**98.99**	57.99	1.65	1.81 (1.79)
17	SVM [26]	SVM (MCV and MCH) (Figure 1)	<0	21.52	**99.93**	73.67	**99.33**	−0.23	0.29 (0.22)

RBC: red blood cells count; Hb: hemoglobin; MCV: mean corpuscular volume; MCH: mean corpuscular hemoglobin; MCHC: mean corpuscular hemoglobin concentration; RDW: red density width; βThal: β thalassemia; α thal: α thalassemia; PPV: positive predictive value; NPV: negative predictive value; SVM: Support Vector Machine. Figures in bold represent results above 90%.

**Table 4 ijms-25-06446-t004:** Comparison of red blood cell count characteristics in the two most common deletions.

α Globin Genetics	αα/-α3.7 kb Mean ± STD (Range)	-α3.7 kb/-α3.7 kb Mean ± STD (Range)	*p* *	αIVS I-1/αα Mean ± STD (Range)	αIVS I-1/αIVS I-1 Mean ± STD (Range)	*p* **	αIVS I-1/ -α3.7 kb Mean ± STD (Range)
Number of individuals (%)	134	24		97	8		7
RBC (×10^9^/dL)	4.8 ± 0.4 (6.16–3.79)	5.3 ± 0.49 (4.37–6.21)	<0.001	4.9 ± 0.36 (4.1–6.1)	5.4 ± 0.57 (4.52–5.96)	0.002	5.5 ± 0.56 (4.72–6.14)
Hb (g/dL)	11.6 ± 0.96 (9.1–14.2)	11.2 ± 1.01 (9.0– 13.3)	NS	11.8 ± 0.88 (9.4–14.8)	10.5 ± 1.19 (9.0–11.7)	<0.001	11.1 ± 0.66 (10.3–11.8)
MCV (fl)	74.4 ± 4.35 (59–83.9)	68.7 ±5.6 (58.7–77.1)	<0.001	73.7 ± 4.36 (53.3– 82.2)	64.9 ± 4.66 (59.1–73.6)	<0.001	63.3 ± 1.41 (61.8–65.2)
MCH (pg)	24.3 ± 1.75 (18–27.1)	21.8 ± 2.11 (17.1–24.6)	<0.001	23.9 ± 1.63 (16.5– 26.5)	19.5 ± 2.15 (17.9–24.7)	<0.001	20.2 ± 0.96 (19.2–21.9)
MCHC (g/dL)	32.6 ± 1.18 (29.7–36.1)	31.7 ± 1.52 (29.2–34.4)	<0.001	32.4 ± 1.2 (28.9–35.7)	30.0 ± 2.2 (25.5–33.5)	<0.001	28.7 ± 6.66 (19.2–35.4)
RDW (%)	14.8 ± 1.65 (12.5–21.7)	15.5 ± 2.56 (13.3–23.4)	NS	14.4 ± 1.29 (12.3– 20.1)	16.3 ± 2.35 (12.6–20.5)	0.001	18.8 ± 2.64 (16.5–22.6)
Hb F (%)	0.5 ± 0.67 (0.1–7.3)	0.4 ± 0.38 (0.2–1.6)	NS	0.5 ± 0.35 (0.1–1.8)	0.6 ± 0.26 (0.3–1.1)	NS	0.5 ± 0.46 (0.2–1.1)
Hb A_2_ (%)	2.7 ± 0.26 (1.7–3.6)	2.6 ± 0.19 (2.2–2.9)	0.09	2.7 ± 0.3 (1.6–3.1)	2.3 ± 0.3 (1.7–2.7)	0.002	2.8 ± 0.15 (2.5–2.9)

SD: standard deviation; RBC: red blood cells count; Hb: hemoglobin; MCV: mean corpuscular volume; MCH: mean corpuscular hemoglobin; MCHC: mean corpuscular hemoglobin concentration; RDW: red density width; Hb F: fetal hemoglobin; Hb A_2_: hemoglobin A_2_. -α3.7: -α3.7 deletion in the α gene; αIVS I-1 (-5 bp) in α_2_ gene. *p* (*)—Comparison between αα/-α3.7 kb and -α3.7 kb/-α3.7 kb. *p* (**)—Comparison between αIVS I-1 (-5 bp)/αα and αIVS I-1(-5 bp)/αIVS I-1 (-5 bp). -α3.7 kb vs. IVS I-1 (-5 bp)—RBC *p* = 0.002—all the other parameters NS. -α3.7 kb/-α3.7 kb vs. αIVS I-1 (-5 bp)/αIVS I-1 (-5 bp) *p* 0.014 for MCH and *p* 0.027 for MCHC. All the other parameters NS. -α3.7 kb/αIVS I-1 (-5 bp) vs. αIVS I-1 (-5 bp)/αIVS I-1 (-5 bp)—Hb A_2_
*p* = 0.005. The other indices NS.

## Data Availability

Data is contained within the article. The whole data presented in this study are available on a local database but the data are not publicly available since in the database the whole data included subjects identifications, identifications that were deleted before performing the analysis presented in the paper.

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
