# Peer review of "Support Vector Machine-Based Formula for Detecting Suspected α Thalassemia Carriers: A Path toward Universal Screening"

_ijms, 2024, doi:10.3390/ijms25126446_

Round 1

Reviewer 1 Report

Comments and Suggestions for Authors

General comments

 The manuscript has been uploaded in Research Gate before the peer review process has been completed.

The type of  manuscript is “ brief report” , in my opinion it is too long to be  considered brief ¡

 Thalassemia major is a severe disease with high morbidity, low quality of life and low life expectancy. Identification of thalassemia carriers is important; as all genetic diseases prevention is important   in the overall management of the disease, reducing  the burden of these disorders on individuals.

 The screening of thalassemia carriers in endemic areas remains a daily challenge for laboratory professionals. Discriminant formulae  based on red cells parameters have been published to aid  in the investigation of microcytic anemia. The indices usefulness is to detect patients with a high probability to be thalassemia carriers, so suspicious samples can be selected for further  analysis, to confirm the presumptive diagnosis of the disease. Those indices have been defined to discriminate Iron deficiency anemia and β-thalassemia, based on red cell parameters, but no specific index has been defined for α-thalassemia screening.

The SVM method t was  published by the authors for detecting blood counts suspected to be from β thalassemia In the present study the method is applied to the detection of alpha carriers in the context of a prevention program for detecting couples at risk of having a child with hemoglobinopathies.

 In the discussion the authors mention that iron status were not analyzed, so IDA was not diagnosed .

 CBC were run on Advia 2120 analyzer, which reports percentages of RBC subsets.

M/H ratio has been recognized as a potent discriminant index  in the meta analysis by Hoffmann et al.  Clin Chem Lab Med 2015; 53(12): 1883–1894.

 Have the authors studied this index in their population ?

 Along the manuscript and in tables 1, 2 and, please use the appropriate decimals and units.

Correct Hgb

 Abstract

It is already established that all published formulas perform significantly better in distinguishing beta  thal

Hoffmann et al.  Critical appraisal of discriminant formulas Clin Chem Lab Med 2017; 55(10): 1582–1591.

 Introduction  

Paragraph 4 lines 62-76 , should be deleted and included in the discussion.

Many statements appear again in the discussion. Delete repetitions

Methods

Authors mention “more than 100,000 subjects were screened and 2,346 pregnancies at risk for having an affected child” ,  could it be possible to know the prevalence of alpha and beta carriers in the area?  

Lines 136-137 “For the data analysis, we included the results of RBC and HPLC analyses of pregnant 136 women or ron women of childbearing age to….”      correct the typo

39 samples were excluded,(  report the % here ) because the  coexistence of thalassemia and iron deficiency could be suspected ,  so in this case the discriminating capacity of the new formula in such condition is not known; this mix condition is common in pregnants, is this an important information missing?

 2.3. The Support vector machine (SVM) algorithm

Report the algorithm formula in this section , not only in the table

 2.5. Data analysis  Parametric tests were used , why?  How normality was assessed?

 Results

 The group defined as α thalassemia carriers included 372 samples that were analyzed  for molecular diagnosis. The results revealed 291 samples with mutations in the α globin gene;   (first group) Normal α globin gene was found in 81 samples (second group). Another 962 samples were defined  suspected α thalassemia carriers but did not undergo molecular analysis (third group)

 So finally the disease was proven on 291 patients ¡   962 patients simply didn´t have a final diagnosis , were the biochemistry results suggestive of IDA or ACD?

 3.2. Analysis of samples suspected to have a diagnosis of Iron Deficiency Anemia

It is not possible to accept “suspected IDA”.    The diagnosis is uncertain, nothing has been proven based in RBC and RDW , read above. Correct the typo  111/962 line 266.

 3.3. Comparison of the results from the two common α globin mutations found.

1337 or 1334?

This section is difficult to read and it is shown in a table, so a brief comment would be  enough , in case of relevant information; in my opinion it is not the case, due to the   few patients in some sub divisions,  the results are not definitive

 Table 1 correct the title of the  row 2 according to text page 5 line 208 “18,572 had blood counts that did not correspond to α or β thalassemia”

Author Response

General comments

 The manuscript has been uploaded in Research Gate before the peer review process has been completed.

The type of  manuscript is “ brief report” , in my opinion it is too long to be  considered brief ¡

Authors answer

Thanks for the comment, the decision to classify the paper as a short report was from the editors of the journal.

 Thalassemia major is a severe disease with high morbidity, low quality of life and low life expectancy. Identification of thalassemia carriers is important; as all genetic diseases prevention is important   in the overall management of the disease, reducing  the burden of these disorders on individuals.

 The screening of thalassemia carriers in endemic areas remains a daily challenge for laboratory professionals. Discriminant formulae  based on red cells parameters have been published to aid  in the investigation of microcytic anemia. The indices usefulness is to detect patients with a high probability to be thalassemia carriers, so suspicious samples can be selected for further  analysis, to confirm the presumptive diagnosis of the disease. Those indices have been defined to discriminate Iron deficiency anemia and β-thalassemia, based on red cell parameters, but no specific index has been defined for α-thalassemia screening.

The SVM method t was  published by the authors for detecting blood counts suspected to be from β thalassemia In the present study the method is applied to the detection of alpha carriers in the context of a prevention program for detecting couples at risk of having a child with hemoglobinopathies.

 In the discussion the authors mention that iron status were not analyzed, so IDA was not diagnosed .

 CBC were run on Advia 2120 analyzer, which reports percentages of RBC subsets.

M/H ratio has been recognized as a potent discriminant index  in the meta analysis by Hoffmann et al.  Clin Chem Lab Med 2015; 53(12): 1883–1894.

Authors response:

Thanks for the comment, we add this meta-anlysis in the references. We agree that the MCV/MCH ration are the best indices.  Have the authors studied this index in their population ? Indeed, our SVM formula indeed use those parameters. A comment was included in the discussion.

 Along the manuscript and in tables 1, 2 and, please use the appropriate decimals and units. All the unit's definition were revised and corrected when applicable.

Correct Hgb – The term was corrected to Hb

 Abstract

It is already established that all published formulas perform significantly better in distinguishing beta  thal

Hoffmann et al.  Critical appraisal of discriminant formulas Clin Chem Lab Med 2017; 55(10): 1582–1591.

Authors response

Thanks for the comment, A citation was added in the introduction.

 Introduction  

Paragraph 4 lines 62-76 , should be deleted and included in the discussion.

Many statements appear again in the discussion. Delete repetitions

Authors response

Thanks for the comment, the reviewer comment may be appropriate, but we choose to include the description of the different forms of alpha thalassemia in the introduction since most of the readers are not familial with those alpha characteristics.

Methods

Authors mention “more than 100,000 subjects were screened and 2,346 pregnancies at risk for having an affected child” ,  could it be possible to know the prevalence of alpha and beta carriers in the area?  

Authors response:

Indeed, it seems to be the possibility of calculating the incidence of alpha and beta thalassemia in our area, but the results may have some bias since not all the pregnant women sent samples for this screening. Beside that the incidence of hemoglobinopathies carriers in our area is not part of the purpose of this study. Then we prefer not to include this data.

Lines 136-137 “For the data analysis, we included the results of RBC and HPLC analyses of pregnant 136 women or ron women of childbearing age to….”      correct the typo

Thanks – the typographic error was corrected.

39 samples were excluded,(  report the % here ) because the  coexistence of thalassemia and iron deficiency could be suspected ,  so in this case the discriminating capacity of the new formula in such condition is not known; this mix condition is common in pregnants, is this an important information missing?

Authors response

The percentage of the 39 excluded samples is 2.9 %, which can be considered insignificant. The excluded samples were from individuals with Hb level below 9 g/dl. The figure was added in the text.  In general, we could include all the data, but we prefer to have a more homogenous population for the data analysis.

 2.3. The Support vector machine (SVM) algorithm

Report the algorithm formula in this section , not only in the table.

Thanks for the suggestion, the formula was included also in the text.

 2.5. Data analysis  Parametric tests were used , why?  How normality was assessed?

 Results

 The group defined as α thalassemia carriers included 372 samples that were analyzed  for molecular diagnosis. The results revealed 291 samples with mutations in the α globin gene;   (first group) Normal α globin gene was found in 81 samples (second group). Another 962 samples were defined  suspected α thalassemia carriers but did not undergo molecular analysis (third group)

 So finally the disease was proven on 291 patients ¡   962 patients simply didn´t have a final diagnosis , were the biochemistry results suggestive of IDA or ACD?

Authors response:

As we describe in the text and in the limitations section the iron status was not part of the analysis performed in the screening scheme.  We agree that this data should be relevant for further analysis but in most of the cited studies this parameter was not included.

 3.2. Analysis of samples suspected to have a diagnosis of Iron Deficiency Anemia

It is not possible to accept “suspected IDA”.    The diagnosis is uncertain, nothing has been proven based in RBC and RDW , read above. Correct the typo  111/962 line 266.

Authors response:

The figure was corrected to 111/962. The definition of “suspected IDA” is as written’ just suspicion based in the accepted increase in RDW as suggesting IDA in comparison to pure thalassemia carriers that shown normal RDW. We will accept any other suggestion for defining this group.

 3.3. Comparison of the results from the two common α globin mutations found.

1337 or 1334? Sorry for the mistake, the correct figure is 1334. The error was corrected.

This section is difficult to read and it is shown in a table, so a brief comment would be  enough , in case of relevant information; in my opinion it is not the case, due to the   few patients in some sub divisions,  the results are not definitive

Authors response:

We agree that all the data is presented in the table (Table 4) then most of the paragraph was deleted.

 Table 1 correct the title of the  row 2 according to text page 5 line 208 “18,572 had blood counts that did not correspond to α or β thalassemia”

Authors response:

Thanks for the comment but the data in the first column is obtained, as described in the text from our previous study, in this study the group of non-carriers was defined as non- beta thalassemia carriers. Then we think that the title is correct and correlate to the data source.

Reviewer 2 Report

Comments and Suggestions for Authors

Dear colleagues,

This article's aim is essential and very relevant nowadays. The authors have done a great job, and it was exciting to read the manuscript. 

Some minor problems:

Most of the problems are regarding punctuation.

1. For citation, use square brackets. All manuscript.

2. Use space before brackets. Lines: 38, 57, 335 (2x), 347, 416.

3. Do not need space after abbreviation pg. Line: 73.

4. Need point after citation. Line: 78, 119, .

5. Check the phrase "ron women." Is it correct? Line: 137.

6. The abbreviation "gr" should be changed to "g". Lines: 138, 370, 372, 375.

7. Do not need a point after the title. Line: 182.

8.  ≤0.05 change to <0.05. Line: 196.

9. Please specify the p-value (between what) in Table 2.   

10. Check the phrase "111 962 samples." Is it correct? Line: 266.

11. Specify what statistical data are presented in Table 4, Mean±SD.

12. Need point after "al." abbreviation. Lines: 352, 365.

13. Unify the abbreviation "vs." or "vs.". All manuscript.

14. Unify 75th and 95th. Lines: 360-363.

Author Response

Comments and Suggestions for Authors

Dear colleagues,

This article's aim is essential and very relevant nowadays. The authors have done a great job, and it was exciting to read the manuscript. 

Dear reviewer

Thanks for your comments, see our responses below for each comment.

Some minor problems:

Most of the problems are regarding punctuation.

  1. For citation, use square brackets. All manuscript. Corrected
  2. Use space before brackets. Lines: 38, 57, 335 (2x), 347, 416.Corected.
  3. Do not need space after abbreviation pg. Line: 73. Corrected.
  4. Need point after citation. Line: 78, 119, . Corrected.
  5. Check the phrase "ron women." Is it correct? Line: 137.

This typing error was corrected.

  1. The abbreviation "gr" should be changed to "g". Lines: 138, 370, 372, 375. Corrected.
  2. Do not need a point after the title. Line: 182. Corrected.
  3. ≤0.05 change to <0.05. Line: 196. Corrected.
  4. Please specify the p-value (between what) in Table 2.

The p value was the multivariate analysis comparing the three parameters, a comment was included in the legend of the table.   

  1. Check the phrase "111 962 samples." Is it correct? Line: 266. The figure was corrected as suggested also from reviewer 1.
  2. Specify what statistical data are presented in Table 4, Mean±SD..The title of the relevant columns was corrected.
  3. Need point after "al." abbreviation. Lines: 352, 365. Corrected.
  4. Unify the abbreviation "vs." or "vs.". All manuscript .Corrected.
  5. Unify 75th and 95th. Lines: 360-363. We prefer to describe both data limits in separate in order to show the significance of the results and the significance of the lower limit also for the 95th percentile.

Reviewer 3 Report

Comments and Suggestions for Authors

The brief report "Support Vector Machine-Based Formula for Detecting Suspected α Thalassemia Carriers: A Path Toward Universal Screening" attracted my attention. The article presents the findings that indicate mathematical formulas that can be used to identify potential α thalassemia carriers. From a general point of view, the manuscript is comprehensive and well-structured, and the language is good, with a few grammatical and syntactic errors. The authors will certainly find them when revising the manuscript.

-Why did the authors decide to use the Support Vector Machine (SVM) algorithm for this study? Why have you not tried using two mathematical formulas to compare outcomes? This point is worth highlighting.

-Based on the results obtained, will this mathematical formula be able to be standardized on an international scale to identify individuals presenting a high risk? Should the different genetic properties of different populations not be taken into consideration, as well as the threshold values for RBC, HgB, MCV and MCH? This should be clarified.

Abstract:

-The abstract should be moderately revised, specifying the originality of the work in the area investigated.

-Please indicate the period and site of compilation for this study.

Introduction:

-An update of the bibliographic data with the addition of recent references from 2024 enhances its quality.

  Methodology:

-A sub-heading entitled Hematological and biochemical analyses should be added to describe the methodology used to perform these analyses in accordance with quality control requirements. Considering that these are the decisive analyses in the context of the study.

- It would be appreciated if the authors could add a flow study summarizing the inclusion and exclusion criteria and specifying the reference population.

-Data analysis: should be more detailed  (For example: How was the standard deviation obtained, etc.?

Abbreviations:

Abbreviations should be revised. They should be indicated in the first indication of the text, including in the abstract. Giving a list of abbreviations after the conclusion would be better.

Unit

Please standardize the units used for some parameters (For example line 372/ 375 gr/dl should be g/dl)

 Other

Line 137:  Please verify: or ron??? women of childbearing age to obtain a more homogeneous population

Line 269: delete the period at the end of the title

Comments on the Quality of English Language

Minor editing of English language required

Author Response

Comments and Suggestions for Authors

Dear reviewer

Thanks for your comments, see our response for each comment.

The brief report "Support Vector Machine-Based Formula for Detecting Suspected α Thalassemia Carriers: A Path Toward Universal Screening" attracted my attention. The article presents the findings that indicate mathematical formulas that can be used to identify potential α thalassemia carriers. From a general point of view, the manuscript is comprehensive and well-structured, and the language is good, with a few grammatical and syntactic errors. The authors will certainly find them when revising the manuscript.

-Why did the authors decide to use the Support Vector Machine (SVM) algorithm for this study? Why have you not tried using two mathematical formulas to compare outcomes? This point is worth highlighting.

Authors response

Thanks for the comment and suggestion. In our study we compared several formulas published in the literature and not only our developed SVM formula. The added value of the SVM formula is that the formula can be trained in any database that may have some subtle differences between different populations in different countries, in different gender and age groups and different cutoffs used.  A comment was added in the conclusions paragraph.

-Based on the results obtained, will this mathematical formula be able to be standardized on an international scale to identify individuals presenting a high risk? Should the different genetic properties of different populations not be taken into consideration, as well as the threshold values for RBC, HgB, MCV and MCH? This should be clarified.

Authors response:

Indeed, the suggested SVM formula can be applied to the routine used counters and show a “flag” in the results indicating that the results are suspicious to belong from a suspected thalassemia carrier, alpha or beta. See paragraph in the conclusions. The purpose to add in the paper analysis of the two common mutations of the alpha gene found in our study was to show the relevance of using our SVM formulaAbstract:

-The abstract should be moderately revised, specifying the originality of the work in the area investigated.

Thanks, we taught that the sentence in the abstract “Based on these results, mathematical formulas incorporating one of these reliable formulas for detecting suspected α or β thalassemia carriers in the program of the automatic analyzers" can flag these results and increase the awareness of the primary physicians about the carrier risk and send an alert with a recommendation for further testing.” shows the importance of this study. Words limitation did not allow us to add another explanation.  

-Please indicate the period and site of compilation for this study. As described in the methods, samples were obtained from 2010 and are part of the prevention and screening program in northern Israel.

Introduction:

-An update of the bibliographic data with the addition of recent references from 2024 enhances its quality.

Thanks, another two new references were added in the bibliography.

  Methodology:

-A sub-heading entitled Hematological and biochemical analyses should be added to describe the methodology used to perform these analyses in accordance with quality control requirements. Considering that these are the decisive analyses in the context of the study.

We add the following subtitle: Read blood count analysis. Biochemical analyses were not done.

- It would be appreciated if the authors could add a flow study summarizing the inclusion and exclusion criteria and specifying the reference population. The flow chart of the screening program that is the source of the data, is presented in the referred paper.  We think that adding more graphs will add much more pages for the paper.

-Data analysis: should be more detailed  (For example: How was the standard deviation obtained, etc.? The data analysis was done using the SPSS standard statistical program. We think that no further description is required.

Abbreviations:

Abbreviations should be revised. They should be indicated in the first indication of the text, including in the abstract. Giving a list of abbreviations after the conclusion would be better. Abbreviations were carefully revised.

Unit

Please standardize the units used for some parameters (For example line 372/ 375 gr/dl should be g/dl) - Corrected and revised all along the manuscript.

 Other

Line 137:  Please verify: or ron??? women of childbearing age to obtain a more homogeneous population. The typing error was corrected.

Line 269: delete the period at the end of the title. Corrected.